# Evaluation of knowledge and awareness of diabetes in higher secondary level students of Kaski district: A cross-sectional study

Arpana Pandit[1], Biswash Sapkota[2], Nishan Poudel[3], Renu Karki[4], Bharat Poudel[5], Ramakanta Lamichhane[6,3]*

1 Department of Biotechnology, School of Science, Kathmandu University, Dhulikhel, Nepal, 2 Department of Pharmacy and Clinical Pharmacology, Madan Bhandari Academy of Health Sciences, Hetauda, Nepal, 3 Gandaki Province Academy of Science and Technology (GPAST), Kaski, Nepal, 4 Pokhara University, Pokhara, Kaski, Nepal, 5 University of Alabama at Birmingham, Birmingham, Alabama, USA, 6 Department of Pharmacy, Kathmandu University, Dhulikhel, Nepal

* ramakanta.lamichhane@ku.edu.np

## Abstract

Studies have shown that the knowledge of diabetes among adults in Nepal is poor. This study aimed to evaluate the knowledge regarding diabetes in secondary level (i.e., Grade 11 and 12) science students. We assessed the student's knowledge and awareness regarding diabetes through self-administered questionnaires consisting of 26 questions associated with symptoms, treatment, prevention, and complications of diabetes. Eight higher secondary schools with science streams in Kaski district were selected. A total of 561 students were in the study. Almost all the students (96%) expressed that they had heard about diabetes but only 37% of students correctly indicated the risk factors of it. One-fourth of the total students were able to distinguish between type 1 and type 2 diabetes. Only 13% of students knew about the third type of diabetes, i.e., gestational diabetes. Nearly 47% of students had a wrong perception that diabetes cannot be prevented or delayed. Though early diagnosis of diabetes is essential to prevent complications, only 61% knew about this fact. In our study, 26% of respondents had a family history of diabetes. Overall, 53% showed poor knowledge, 44% showed average knowledge and only 3% showed good knowledge regarding diabetes. The study showed that the gender of the participants and the type of college (government or private) they were studying had no significant relationship with the knowledge level regarding diabetes. However, family history and educational level (class 11 or 12) showed a positive relationship with the level of knowledge regarding diabetes. Students had good knowledge regarding the symptoms of diabetes but had poor knowledge about risk factors, complications, and prevention strategies for diabetes. Overall diabetes knowledge was inadequate in the students, indicating that there is a necessity for urgent action to enhance students' knowledge regarding diabetes.

**Data availability statement:** All relevant data are within the manuscript and its Supporting information files.

**Funding:** The author(s) received no specific funding for this work.

**Competing interests:** The authors have declared that no competing interests exist.

## Introduction

Diabetes is defined as an increased glucose level in the blood due to a lack of insulin in our body and/or insulin resistance. Insulin is a hormone that metabolizes the glucose in our body. The metabolism of glucose is affected due to a lack of insulin or its impaired function which raises the blood glucose level, and causes diabetes. The risk factors for diabetes include obesity, hypertension, aging, unhealthy/poor diet, lack of exercise, etc. Diabetes is classified into three types: Type 1 diabetes (T1D), Type 2 diabetes (T2D), and gestational diabetes. Around 90 to 95% people suffering from diabetes is associated with T2D.

The prevalence of T2D in Nepal is high. Only taking medicines is not sufficient to control this disease. Proper diet plan, lifestyle modification, and exercise are equally important. In addition, awareness and proper knowledge are very helpful and necessary to prevent or delay this disease. Furthermore, it will help to improve the quality of life of patients suffering from diabetes. With the knowledge about this disease among the present young and adult population we are making our future elderly population aware of diabetes. There have been various studies related to the assessment of knowledge of diabetes in patients and health care providers [1–3] In Nepal, many research works regarding the knowledge and awareness of patients and health care providers have been conducted. The data show that less than 25% of diabetes patients had highly satisfactory knowledge regarding diabetes [2,4]. Studies also have shown that a large proportion of non-diabetes patients had poor diabetes knowledge [5]. In a study conducted on health care professionals, knowledge regarding various aspects of diabetes in the preclinical medical student indicated that a few students have a better knowledge regarding the disease [6]. The poor knowledge of diabetes patients, non-diabetes patients and graduate students demanded some urgent need to find the possible drawbacks or lacking.

In the current study, we evaluated the awareness level regarding diabetes among higher secondary students who are just starting their adulthood. So, far no studies have been conducted in Nepal to assess the knowledge of diabetes among higher secondary-level students. In this cross-sectional study, we will assess the student's knowledge and awareness of disease-specific information, symptoms, treatment, prevention and complications of diabetes [7]. From this study, we expect to know their level of awareness regarding the disease and alert the responsible parties and authorities to work to enhance the level of knowledge regarding diabetes. This study will also be helpful for the government, policymakers, and higher secondary education board to take necessary steps for enhancing the knowledge and awareness among the higher secondary graduating students.

## Methodology

### Study design

We conducted a descriptive cross-sectional study among the science students of secondary education (Grade 11 and 12) studying in different educational institutes in Kaski district to evaluate their knowledge and awareness regarding the various aspects of diabetes. We aimed to determine the influence of diabetic history, gender, and education level on the knowledge and awareness of diabetes. We also tried to evaluate if there is any difference in the level of knowledge and awareness regarding diabetes between the students of community (government) colleges and private colleges. The data were collected from October 2022 through December 2023.

### Study setting and sample size

The primary reason for selecting science students of grades 11 and 12 is that these are the last two grades in school education in Nepal after which students go to universities for specific subjects and they will not get a chance to study health education and diseases (except pursuing

medical and paramedical subjects) [8]. Therefore, assessing their knowledge about diabetes would provide an estimation of how effective the health educational programs during their schooling were. The second reason is that they would soon turn into adults, so the knowledge and awareness of diabetes would be important to prevent them from getting diabetes in their adulthood. We hypothesize that science students would have higher knowledge and awareness regarding diabetes compared to their non-science counterparts.

The secondary education of science (Grades 11 and 12) in Nepal is provided through community (government) based or private (boarding) secondary schools/colleges and according to the data of 2021 provided by the National Examination Board (NEB), Pokhara, 47 higher secondary schools in Kaski district offered a science stream. Of them, 13 were community or government colleges and the remaining 34 were private colleges. We divided the 47 colleges into two strata – Community/government college (13) and private college (34). We selected 4 colleges randomly from each stratum so that our total sample number for colleges would be eight. According to the data provided from NEB, Pokhara, there are about 5500 students currently (2021) studying in the science stream in those 47 colleges. We took 561 students as the sample which was around 10% of the total population.

## Study tool

We adopted the previously developed questionnaire with certain modifications to evaluate the knowledge regarding diabetes along with other important aspects of the disease like prevention, management, and complications. We also looked into the distribution of the family history of diabetes in participating students along with the age group of the diabetic family member. The questionnaires consisted of 27 questions (English with Nepali translation) regarding different aspects of diabetes [9]. The questions were designed after reviewing different research articles and consulting with endocrinologists affiliated with authorized institutions/hospitals [10–12]. The questions were revised based on a pilot study conducted with 50 students to eliminate any confusion or misinterpretation. The responses from the pilot study were used to enhance the questionnaire but were not included in the final results. The questions were made as simple as possible so that all the students would understand properly.

## Data collection and participation

We obtained a letter of permission from each college to survey the research work. After approaching the college, science students (class 11 or 12) were identified and explained to them our objective and research study. All students were consented to take part in the study. They were assured verbally and in writing that their personal information would be kept strictly confidential and only used for research purposes. A written consent was obtained from each participant. Students below the age of 18 were further asked for written consent from their parents by getting an authorized signature on the parent consent form, from their parents. The questionnaire is self-administered. Participants were instructed to seek clarification if they found any part of the content difficult to understand. Ethical approval to conduct this study was obtained from the Nepal Health Research Council (Registration number: 213/2022 P).

## Data analysis

The data were entered in Excel and analyzed using SPSS version 24. For analyzing the knowledge level, scores of 1 were given to correct and 0 to the wrong answers respectively. The maximum score would be 26. The percentage of the total score for each participant was evaluated as:(1)

$$Total\ score\ (\%) = \frac{Correct\ answers\ (total\ score)}{26} \times 100\% \tag{1}$$

Based on the score obtained, the students were divided into three categories: good knowledge if they correctly answered 21 or more questions (>80%), average knowledge if they correctly answered 16–20 questions (60–80%), and poor knowledge if they correctly answered less than 16 questions (<60%). After that, the percentage of participants who fell into each of the categories was determined. The percentage of students who knew or didn't know about the particular aspect of the disease was also calculated separately.

Microsoft Excel was used to compute descriptive statistics such as percentage, mean and standard deviation. To find any association between family history of diabetes (presence of diabetic family member), gender, and education level of the students with the knowledge level of diabetes, a test of significance was performed using chi-squared test. *P < 0.05* was considered statistically significant.

## Results

### Demographics of participating students

Around 561 students were enrolled in the study among which 49% (n = 277) were male and 51% (n = 284) were female (Table 1). About 60% (n = 335) of the total students were 16 or 17 years old, while the remaining 40% (n = 226) were of age 18 and above. More than half of the students (58%) were from class 12. The mean age ± SD of age of the participant was 17.3 ± 0.8.

### Knowledge level in different aspects of diabetes

The knowledge distribution of students regarding diabetes is shown in Table 2. Each student was given a set of questionnaires containing 26 questions regarding different aspects of diabetes like: specific information, signs and symptoms, treatment, prevention, and complications.

**Table 1. Demographic characteristics of participating students (N = 561).**

| Characteristics | Number (%) |
|---|---|
| Sex | |
| Male | 277 (49) |
| Female | 284 (51) |
| Age, years | |
| Below 18 | 335 (60) |
| 18 and above | 226 (40) |
| High school education | |
| Class 11 | 234 (42) |
| Class 12 | 327 (58) |
| Family history of diabetes | |
| Yes | 144 (26) |
| No | 417 (74) |
| Age group of family history of diabetes, years | |
| 30–40 | 17 (12) |
| 40–60 | 74 (74) |
| 60–80 | 53 (16) |

**Table 2.  Knowledge regarding diabetes along with number of students who answered correctly (N) and their percentage.**

| Knowledge regarding symptoms/treatment of DM | N (%) |
|---|---|
| What are the common symptoms of diabetes? | |
| *Excessive thirst* | 410 (73) |
| *Excessive urination* | 512 (91) |
| Does wound healing take a long time in diabetes? | |
| *Yes* | 448 (80) |
| Do diabetes patients need to visit an eye doctor? | |
| *Yes* | 215 (38) |
| What are the appropriate treatment methods for diabetes? | |
| *Medication, control diet, and regular exercise* | 521 (93) |
| **Knowledge regarding complications** | **N (%)** |
| Can diabetes cause blindness? | |
| *Yes* | 237 (42) |
| Can diabetes lead to kidney failure? | |
| *Yes* | 294 (52) |
| Can early detection prevent diabetes complications? | |
| *Yes* | 347 (62) |
| **Knowledge regarding risk factors of diabetes** | **N (%)** |
| Which age group has a higher risk/chance of diabetes? | |
| *More than 35* | 414 (74) |
| Can diabetes be prevented? | |
| *Yes* | 293 (52) |
| Can diabetes be delayed? | |
| *Yes* | 307 (55) |
| What are the preventive measures for diabetes? | |
| *Control diet and regular exercise* | 373 (66) |
| Do you know the **risk factors** of diabetes? | |
| *High carbohydrate/sugar-containing diet* | 174 (30) |
| *Lack of exercise* | 14 (2.5) |
| *Obesity* | 14 (2.5) |
| *Heredity/genetics* | 4 (1) |
| *High BP* | 11 (2.5) |

Almost all of the students (96%) expressed that they had heard the term diabetes and also were aware (94%) that it is non-communicable. More than half (60%) believed that the disease was associated with an organ, i.e., pancreas. Around 40% had the misconception that diabetes was associated with other organs like liver, heart, and kidney. However, the majority of them (93%) were aware that the major hormone playing a role in diabetes was insulin. Likewise, 85% of the students knew the prominent feature of diabetes, i.e., increase in the level of glucose in the blood. However, many of them (62%) were unaware of the standard level of glucose (fasting blood glucose level) they should maintain to prevent diabetes.

One-fourth of the total students were aware of two types of major diabetes, i.e., T1D and T2D. Only 13% of students revealed another third type of diabetes, i.e., gestational diabetes. This indicated that very few students knew about gestational diabetes. Although, a high number of students knew about two types of diabetes, very few (32%) were able to distinguish the most common form of diabetes, i.e., T2D diabetes. Around one-fourth had the idea that an age

group of more than 35 had a higher risk for diabetes. Around half of the students had a wrong perception that diabetes cannot be prevented and delayed. The number of students knowing about the risk factors of diabetes was also discouraging. Only 38% of students correctly indicated the risk factors of diabetes. The risk factor of diabetes that the majority of students (30%) mentioned was the consumption of a high carbohydrate or sugar-containing diet. Very few students knew that obesity (2.5%), heredity/genetics (below 1%), high blood pressure (2.5%) etc. are also risk factors for diabetes. The majority of students (85%) were able to identify the classic symptoms of diabetes such as frequent urination, frequent thirst, and long wound healing time.

Most of the students (93%) knew that diabetes patients should take care of diet and exercise regularly apart from taking medicine for proper management of the disease. In addition, 95% of students were aware that diabetics need a special diet. However, when asked about the type of diet, only 50% of the students knew that these patients should not eat high carbohydrate or sugar-containing food. Only 66% knew about the management of prediabetic conditions which strongly require regular exercise and diet control rather than using medication. Half of the students were not aware of the complications of diabetes like blindness and kidney failure. Only 42% of students knew that diabetes can be diagnosed with the measurement of fasting blood glucose (FBG) and postprandial blood glucose (PPBG) levels. Only 61% knew about the fact that early diagnosis of diabetes is crucial to prevent complications. The number of students who knew the importance of regular eye checkups for diabetes patients was as low as 40%.

Around 144 students (26%) responded that they had a family member suffering from diabetes in their family. The ages of diabetic family members mentioned by the students ranged from as low as 34 years to as high as 80 years. Among the students who had diabetic family members, 17% had diabetic family members in the young and early age of 30–40.

## Overall knowledge regarding diabetes and its association with other variables

The overall knowledge score of each student was evaluated from the total marks obtained for each correct answer for 26 questions (n = 26). The mean ± SD diabetes knowledge score (out of 26) of the respondents was 16.5 ± 3.1. Based on the score percentage obtained by the student, we categorized the knowledge level into three categories: good knowledge (>80%), average knowledge (60–80%), and poor knowledge (<60%). We observed that of the whole population, 53% showed poor knowledge, 44% showed average knowledge and only 3% showed good knowledge.

Table 3 shows the results of the analysis of the relationship between diabetes knowledge level and the different characteristics of the participating students. According to the results, no significant difference was observed in the score of knowledge regarding diabetes between male and female respondents (P value 0.74). It indicated from our study that there is no association of knowledge level with the gender of the respondents. However, the grade of the students (class 11 or 12) in which they were studying was significantly associated with the knowledge level. The students from higher grades, i.e., class 12, showed higher mean knowledge scores. We found no such significant relationship existed between the knowledge level of diabetes and the type of college the students were studying. The study showed a significant association between the family history of diabetes and the knowledge level of respondents (P value 0.025). Students who had diabetes patients at home showed a higher mean knowledge score regarding diabetes.

## Discussion

Diabetes has become a major health concern in the countries of South Asia. Based on various studies, it has been estimated that the prevalence of diabetes (T2D) may increase by over 15%

**Table 3. Correlation of characteristics of the participants with mean knowledge score.**

| Dependent Variable (characteristics of the participants) | Mean Knowledge Score (%) | P-value |
|---|---|---|
| **Gender** | | |
| Male | 59.3 | 0.74 |
| Female | 59.7 | |
| **High school education** | | |
| Class 11 | 58.0 | 0.01 |
| Class 12 | 60.0 | |
| **College** | | |
| Government | 59.5 | 0.92 |
| Non-government | 59.5 | |
| **Family history** | | |
| **Respondent having no family history of diabetes** | 58.8 | 0.025 |
| **Respondent having family history of diabetes** | 61.4 | |

Results were obtained from chi-squared tests. P < 0.05 is considered statistically significant.

between 2000 and 2030. Furthermore, the distressing scenario is that South Asian patients with diabetes (T2D) are younger compared to other areas. In addition to it, the progression of diabetes is also found to be more rapid among South Asians [13]. Nepal, a country in South Asia has also shown an increasing tendency of prevalence of T2D as it increased from 8.4% in 2014 to 8.5% in 2020 [14]. The international diabetes federation (IDF) has also emphasized the dire need for public awareness in Nepal regarding diabetes to prevent diseases and complications [15].

Various studies have been done in Nepal to assess the knowledge and awareness regarding T2D in different groups of the population. A study done in a hospital in Bhaktapur showed that the visitors lacked appreciable knowledge and awareness regarding the risk factors and preventive measures of diabetes [16]. Similarly, a study done in Pokhara municipality in Gandaki province showed medium awareness regarding diabetes in diabetic patients [2]. The study also revealed the high prevalence of T2D (11.7%) in Gandaki province which was greater than the overall prevalence in Nepal which is 6.3% (IDF, 2022). In some major cities like Kathmandu, the prevalence of T2D is as high as 15–22% as reported by the world diabetes foundation (WDF). A study done in the National Ayurveda Research and Training Center, Kirtipur to assess the level of diabetes knowledge also revealed a very poor level of diabetes knowledge among diabetes patients in Nepalese adults [17]. All the studies done on diabetic patients in Nepal indicate an urgent need for awareness among the adult generations. It cannot be denied that the high school students at present are the future adult generation. So, their diabetes knowledge determines the awareness of future adult generations of diabetes. This correlation strongly emphasizes the need to evaluate the knowledge of the young generation regarding diabetes to ensure that future adult generations are more aware of the chronic disease. The increasing incidence of diabetes in younger age groups also demands the evaluation of diabetes knowledge in the younger generation. All these backgrounds motivated us for this study in which the young generation is the target population. We tried to evaluate the knowledge level of diabetes among high school students of the science stream. To the best of our knowledge, this type of cross-sectional study on higher secondary students (class 11 and 12) of science background is for the first time in Nepal. Our study showed an average level of knowledge in half of the respondents while a good knowledge level only in 3% population. Such a

discouraging scenario of diabetes knowledge in students of the science stream may provide a basis to predict that the situation would be more depressing in students from non-science background even though it may be a part of another study. Similar to our results, a study done on young students of Grades 9 and 10 in schools of Bharatpur metropolitan city and Bidur municipality found that the majority of students showed poor knowledge [18]. The study also revealed the data that about 21% of the respondents had a family history of diabetes which was as high as found in our study (26%). Though both studies were done in different places in Nepal, the results of a higher percentage of family history of diabetes indicate a higher prevalence of diabetes in Nepal.

A sedentary lifestyle and less physical exercise play a negative role in developing T2D. Obesity is also highly linked with T2D. Heredity or family history of diabetes increases the chance of diabetes in the offspring. Various studies done inside or outside Nepal have shown a good association between high BP and T2D, indicating a higher risk of diabetes in patients with high BP [2]. We showed that students had good knowledge about the symptoms of diabetes but lacked knowledge regarding the risk factors of diabetes. This sort of ignorance of students about the importance of physical exercise, controlling body weight, and attention to family history would increase the prevalence and complications of diabetes. Hence, it seems necessary to promote physical exercise to them along with counseling on the risk factors of diabetes. The students should also be motivated to take part in the care of diabetic patients in the family and community which will help them to make them more aware regarding the disease.

Gestational diabetes mellitus (GDM), a type of diabetes in women during the time of pregnancy, is emerging as a serious issue worldwide due to its increasing number of cases. Nepal is also not away from this problem. Various studies in different hospitals and places have shown a high prevalence of GDM from 1.58 (2014) % to 4.5%. (2017) [19,20]. Studies have shown that women with previous GDM have a substantially higher risk of developing DM and cardiovascular disease. Similarly, the offspring of mothers with GDB are highly prone to suffer from adverse outcomes such as hypoglycemia, and DM later in life [21]. Proper diagnosis and treatment at the right time (as early as possible) and raising awareness from the grassroots level have been suggested by various scholars and researchers to control the increasing burden of GDM [22]. In our study, very few (13%) students were aware of GDM which is a sign of a hard situation in the future to control GDM.

The correlations study showed no association of gender and the type of college (government or private) the student being studied with the level of knowledge (score) regarding diabetes. However, we found a significant association between knowledge level regarding diabetes with the education level (grade 11 or 12) and family history of diabetes. Students in grade 12 showed higher scores of knowledge compared to grade 11. It was consistent with other studies done on medical students, in which students who completed a greater number of years in higher education had higher levels of knowledge regarding diabetes [6,23]. This may be due to the addition of knowledge from the additional curriculum and courses in grade 12. Our study showed a positive relationship between knowledge level and family history of diabetes. The students who had a family history of diabetes had significantly higher knowledge scores compared to those not having a family history of diabetes. Such types of results have been reported in other studies also [24]. The student might get a chance to get more information from the diabetic family members which may help to increase their knowledge level regarding diabetes. It also depends upon how the students provide help or support to the diabetic family members. More interaction, care, and support to the diabetic family members may enhance their knowledge level regarding diabetes.

Knowledge is the critical factor that motivates to bring behavioral changes to an individual. Once awareness is created or raised, people are more likely to take action or work on prevention

and control activities regarding a particular disease. The students were found to have a low and inadequate level of knowledge and awareness regarding diabetes. While some of them possessed various misconceptions about this particular chronic disease. Health and educational authorities at the government level and school authorities in the region should focus on suitable and possible strategies required to enhance the level of knowledge and awareness of the students regarding diabetes. Efforts from the national level and interventions focused on health-related school education would help to increase awareness about diabetes among the students.

## Conclusion

Students included in our study had good knowledge regarding the symptoms of diabetes but had poor knowledge about its risk factors, complications, and prevention strategies. Overall, we see that the students were found to have a low and inadequate level of knowledge and awareness regarding diabetes. Health and educational authorities at the government level and school authorities in the region should focus on suitable and possible strategies to enhance the level of knowledge and awareness of the students regarding diabetes. Efforts from the national level to incorporate essential materials in the course of study and curriculum regarding diabetes is necessary. Interventions focused on health-related school education would also help to increase awareness about diabetes among the students.

Education of these students about diabetes will help to increase the quality life of the patient in the family and community and even help themselves to prevent and delay this disease in the future. They can influence the community directly and their knowledge about the disease will determine the health of the next generation. The greater emphasis on strategies for continuous education programs regarding the disease will also help to control the increasing incidence and prevalence of diabetes globally.

## Limitations and recommendations

The study may have a few limitations that might affect the validity of the results we obtained. In this study, we could not include the students of other fields. However, it is assumed that students from the science stream have better knowledge regarding diseases. Due to the cross-sectional nature of the study, we could not determine the temporal relationship of our findings. Also, as respondents had responses self-administered, there could have been information bias. As our study showed poor knowledge regarding diabetes in the students of the science stream, we recommend that future research be conducted among the students of other non-science streams to evaluate the knowledge of diabetes.

## Supporting Information

**S1 Table. Data Collected from each participant.**
(XLSX)

**S1 Text. Questionnaire in English and Nepali language.**
(DOCX)

## Acknowledgment

We are thankful to all the colleges who permitted us to conduct the research work. We are also thankful to the National Examination Board (NEB), Pokhara for providing the information regarding the colleges that provide higher secondary education and the number of students in the science stream. We express our kind acknowledgment to the Nepal Health Research Council for critical review and ethical clearance regarding our proposal.

## Author contributions

**Conceptualization:** Ramakanta Lamichhane.

**Data curation:** Biswash Sapkota, Nishan Poudel, Renu Karki, Ramakanta Lamichhane.

**Formal analysis:** Biswash Sapkota, Nishan Poudel, Renu Karki, Bharat Poudel.

**Investigation:** Arpana Pandit, Nishan Poudel, Bharat Poudel, Ramakanta Lamichhane.

**Methodology:** Arpana Pandit, Biswash Sapkota, Nishan Poudel, Renu Karki, Bharat Poudel, Ramakanta Lamichhane.

**Supervision:** Bharat Poudel, Ramakanta Lamichhane.

**Validation:** Ramakanta Lamichhane.

**Writing – original draft:** Ramakanta Lamichhane.

**Writing – review & editing:** Arpana Pandit, Biswash Sapkota.

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
