## [Decision Letter · Decision Letter 0]

13 Aug 2024

Dear Dr. Lamichhane,

Thank you for submitting your manuscript to PLOS ONE. After careful consideration, we feel that it has merit but does not fully meet PLOS ONE’s publication criteria as it currently stands. Therefore, we invite you to submit a revised version of the manuscript that addresses the points raised during the review process.

We look forward to receiving your revised manuscript.

Kind regards,

Sunil Shrestha

Academic Editor

PLOS ONE

Journal Requirements:

2. We note that your Data Availability Statement is currently as follows: "All relevant data are within the manuscript and its Supporting Information files."

Additional Editor Comments:

Major Revision

Reviewers' comments:

Reviewer's Responses to Questions

**Comments to the Author**

1. Is the manuscript technically sound, and do the data support the conclusions?

Reviewer #1: Yes

2. Has the statistical analysis been performed appropriately and rigorously?

Reviewer #1: No

3. Have the authors made all data underlying the findings in their manuscript fully available?

Reviewer #1: Yes

4. Is the manuscript presented in an intelligible fashion and written in standard English?

Reviewer #1: Yes

Reviewer #1: The current study assesses the knowledge and awareness of diabetes in higher secondary-level students. I have some questions and suggestions given below:

1. In the introduction section, the author described that most of the population had poor knowledge including health professionals. This kind of background makes it rational for intervention studies instead of observational studies. Previously it has been reported that education has a significant impact on diabetes knowledge, which means higher education has higher diabetes knowledge. There it is suggested that kindly provide a proper rationale for the current study.

2. In the method section, what is the rationale for taking 10% of the population, and how did the authors conclude, that 10% of the population is representative of the total population?

3. What is the need for the development of the study tool for the current study, if already developed tools are available like the Diabetes Knowledge Test?

4. Kindly provide the pilot study result in the method section and clarify whether the pilot study participants were included in the final analysis.

5. Was the use self-administered questionnaire?

6. In the result section, it would be great if you stratified the results based on the student grades.

7. In Table 3, the author used the word “correlation” which is specifically used for the correlational analysis in the statistics, however, the author used a paired t-test. I am wondering why the author used a paired t-test instead of using independent t-test.

8. In discussion, “the international diabetes federation (IDF) has also emphasized on the dire need of public awareness in Nepal regarding diabetes to prevent the diseases and the complications.” Kindly look at this statement, is that correct, if yes kindly provide the reference.

9. In the discussion, the author discusses the knowledge of the participants in the hospital setting, however, the study population of the current study is students of higher education, what is the rationale of comparing these students with other study settings?

10. In general, kindly look for grammatical mistakes and look at the sentence structure in terms of its meaning concerning your current study.

**Do you want your identity to be public for this peer review?** For information about this choice, including consent withdrawal, please see our Privacy Policy

Reviewer #1: No

---

## [Author Response · Author response to Decision Letter 1]

19 Oct 2024

Responses to the Comments from Reviewer:

1. In the introduction section, the author described that most of the population had poor knowledge including health professionals. This kind of background makes it rational for intervention studies instead of observational studies. Previously it has been reported that education has a significant impact on diabetes knowledge, which means higher education has higher diabetes knowledge. There it is suggested that kindly provide a proper rationale for the current study.

Answer: Thank you for your comment. In the introduction part we have depicted the poor knowledge of healthcare professionals like medical students during their preclinical courses (i.e. 1st and 2nd year) based on some studies. At this stage, their knowledge level is also affected by their education system and exposure during their studies in higher secondary level which is just before the start of the journey of medical science studies. So, the poor knowledge level of diabetes knowledge of preclinical medical students may also indicates their poor knowledge of higher secondary students which can be proved after we extensively researched diabetes knowledge levels in students of higher secondary levels. So, such a scenario was presented in the introduction part.

The reviewer has suggested the necessity of intervention studies instead of observational studies as we have many studies that reflect poor diabetes knowledge in most of the population. We really acknowledge this opinion from the reviewer. However, in Nepal, no studies have been conducted in higher secondary students regarding the knowledge of diabetes. Though we can take reference from other countries, but we felt that it would be better to find out the actual scenario in our Nepalese community rather than generalizing from other studies. For that concern we opted for this study. We believe that the results of this study will motivate for the interventional studies in future

2. In the method section, what is the rationale for taking 10% of the population, and how did the authors conclude, that 10% of the population is representative of the total population?

Answer = It has been accepted that a good maximum sample size would be usually 10% of the total population as long as it does not exceed 1000 (https://survicate.com/blog/survey-sample-size/). In our case, total population was 5500 (students in the science stream) and its 10% would be 550. So, we took 561 (near to 550) students as the sample for the study.

We could have calculated the sample size using the Rao soft sample size calculator (http://www.raosoft.com/samplesize.html.) with a 95% confidence level, a 5% margin of error and assuming the response distribution for each question to be 50%. The calculated sample size would be 377 individuals.

Though the two methods are equally acceptable, we opted to survey for higher number of samples i.e. 561 individuals to increase the dependability of the results.

3. What is the need for the development of the study tool for the current study, if already developed tools are available like the Diabetes Knowledge Test?

The target group for this research is Higher secondary students. They have different education levels and backgrounds compared to lay people. Therefore, we modified the questionnaire and developed our own. The questionnaire is attached here in the appendix below.

4. Kindly provide the pilot study result in the method section and clarify whether the pilot study participants were included in the final analysis.

Answer= The questions were revised based on a pilot study conducted with 50 students to eliminate any confusion or misinterpretation. The tables given below include results regarding the pilot study. From the pilot study we found insufficient knowledge levels in students regarding diabetes.

Table: Demographic characteristics of participating students (N = 50)

Characteristics Number (%)

Sex

Male 27(54)

Female 23 (46)

High school education

Class 11 28(56)

Class 12 22 (44)

Table 3: Correlation of characteristics of the participants with mean knowledge score

Dependent Variable Mean Knowledge Score (%) P-value

Gender

Male 51.2 0.61

Female 54.1

High school education

Class 11 52.0 0.03

Class 12 60.0

Results were obtained from chi-squared tests. P-value <0.05 is considered statistically significant.

The responses from the pilot study were used to enhance the questionnaire but were not included in the final results. Considering the word limit of the journal for the manuscript, we decided to exclude the content of the pilot study.

5. Was the use self-administered questionnaire?

Answer=The questionnaire is self-administered. Participants were instructed to seek clarification if they found any part of the content difficult to understand. Please refer to the comment 3 above.

6. In the result section, it would be great if you stratified the results based on the student grades.

We equally agree that it would have been much better if we had stratified the results based on the student grades. It would help us to find some correlation between grades and the knowledge level of students. However, during the time of the survey, we didn’t include the requirement to write the grades for the students while filling out the questionnaire. So, we couldn’t collect any data related to student grades. So, in this study, we could not include the results based on the students’ grades. However, this could be an important point to consider in future studies. Thank you for raising such an important issue which we missed it at the initial time of the study.

7. In Table 3, the author used the word “correlation” which is specifically used for the correlational analysis in the statistics, however, the author used a paired t-test. I am wondering why the author used a paired t-test instead of using independent t-test

Answer= The test was done using chi-squared test instead to paired t-test, which was a typo on our hand. Throughout the manuscript, we replaced paired t-test with chi-squared test.

8. In discussion, “the international diabetes federation (IDF) has also emphasized on the dire need of public awareness in Nepal regarding diabetes to prevent the diseases and the complications.” Kindly look at this statement, is that correct, if yes kindly provide the reference.

Answer: Thank you for notifying us to add the reference to a statement that was missed. The new reference is included in the bibliography (Ref. no. 15).

9. In the discussion, the author discusses the knowledge of the participants in the hospital setting, however, the study population of the current study is students of higher education, what is the rationale of comparing these students with other study settings?

This is the first study done in Nepal regarding high school students. We didn’t find any study to compare the knowledge of diabetes among higher secondary students in Nepal. So, for comparative study, we took studies focused on medical students (1st/2nd year) in a hospital setting as a reference.

10. In general, kindly look for grammatical mistakes and look at the sentence structure in terms of its meaning concerning your current study.

Thank you so much for your time and valuable suggestions and comments. We have a strong belief that your comments and guidance would certainly enhance the quality of the article. We have made many grammatical corrections as per your suggestion after thorough investigations. We hope that we have tried our level best to satisfy your queries.

---

## [Decision Letter · Decision Letter 1]

31 Oct 2024

Evaluation of knowledge and awareness of diabetes in higher secondary level students of Kaski district: A cross-sectional study

PONE-D-24-24606R1

Dear Dr. Lamichhane,

We’re pleased to inform you that your manuscript has been judged scientifically suitable for publication and will be formally accepted for publication once it meets all outstanding technical requirements.

Kind regards,

Wen-Jun Tu

Academic Editor

PLOS ONE

Additional Editor Comments (optional):

Reviewers' comments:

Reviewer's Responses to Questions

**Comments to the Author**

Reviewer #1: All comments have been addressed

2. Is the manuscript technically sound, and do the data support the conclusions?

Reviewer #1: Yes

3. Has the statistical analysis been performed appropriately and rigorously?

Reviewer #1: Yes

4. Have the authors made all data underlying the findings in their manuscript fully available?

Reviewer #1: Yes

5. Is the manuscript presented in an intelligible fashion and written in standard English?

Reviewer #1: Yes

Reviewer #1: (No Response)

**Do you want your identity to be public for this peer review?** For information about this choice, including consent withdrawal, please see our Privacy Policy

Reviewer #1: No

---

## [Editor Report · Acceptance letter]

PONE-D-24-24606R1

PLOS ONE

Dear Dr. Lamichhane,

I'm pleased to inform you that your manuscript has been deemed suitable for publication in PLOS ONE. Congratulations! Your manuscript is now being handed over to our production team.

Kind regards,

on behalf of

Dr. Wen-Jun Tu

Academic Editor

PLOS ONE